# Integration of iPSC-Derived Microglia into Brain Organoids for Neurological Research

**DOI:** 10.3390/ijms25063148

**Published:** 2024-03-09

**Authors:** Muhammad Asif Mrza, Jitian He, Youwei Wang

**Affiliations:** Institute of Medical Engineering & Translational Medicine, Tianjin University, Tianjin 300072, China; mirzasif678@gmail.com (M.A.M.); a1063415466_@tju.edu.cn (J.H.)

**Keywords:** immunized organoids, iPSCs, microglia, neurons, Alzheimer’s disease

## Abstract

The advent of Induced Pluripotent Stem Cells (iPSCs) has revolutionized neuroscience research. This groundbreaking innovation has facilitated the development of three-dimensional (3D) neural organoids, which closely mimicked the intricate structure and diverse functions of the human brain, providing an unprecedented platform for the in-depth study and understanding of neurological phenomena. However, these organoids lack key components of the neural microenvironment, particularly immune cells like microglia, thereby limiting their applicability in neuroinflammation research. Recent advancements focused on addressing this gap by integrating iPSC-derived microglia into neural organoids, thereby creating an immunized microenvironment that more accurately reflects human central neural tissue. This review explores the latest developments in this field, emphasizing the interaction between microglia and neurons within immunized neural organoids and highlights how this integrated approach not only enhances our understanding of neuroinflammatory processes but also opens new avenues in regenerative medicine.

## 1. Introduction

Induced Pluripotent Stem Cells (iPSCs), first generated in 2006 [1], reprogram mature somatic cells, such as skin or blood cells, back into a pluripotent state. These reprogrammed cells can then be differentiated into various specific cell lineages [2]. This versatility underscores the importance of iPSCs in both medical research about the disease pathogenesis of and clinical therapeutic applications [3,4].

Neurodegenerative diseases, which progressively and selectively deteriorate specific neuron populations, represent a growing medical challenge in an aging global population [5]. Most current knowledge about the pathology and mechanisms of neurodegenerative diseases stems from histopathological studies. However, obtaining brain samples from patients is challenging [6,7]. While animal models offer valuable insights in neurological research, they fail to completely replicate human neural disease phenotypes [8,9]. A notable difference is in the neuron structure; for instance, human dopamine neurons form about 1–2.5 million symmetrical synapses, with an average total length of 4.6 m, which is ten times more than that of rats [10]. Additionally, the production and development of neurons significantly differ between humans and mice. For example, human interneurons are generated from dorsal progenitors, a process not observed in mice [11,12]. The unique neurogenesis and neurodevelopment processes in humans lead to complex neuronal networks that cannot be adequately studied using animal models. In light of this, the utilization of iPSCs presents a valuable approach to surmount the existing disparities in neurological research [13].

Microglia develop from erythro-myeloid precursors located in the yolk sac during embryonic development and subsequently take residence in the central nervous system [14]. Differing from this, monocytes that originate in the bone marrow, can also travel to the central nervous system and subsequently differentiate into macrophages. Some surface markers, such as Tmem119, help differentiate microglia within the central nervous system [15]. In some diseases, microglia and recruited macrophages may play different pathological roles. For example, in multiple sclerosis (MS), active multiple sclerosis lesions primarily consist of microglia, and as the lesions mature, the number of microglia gradually decreases, while the quantity of macrophages recruited from peripheral monocytes increases [16]. Inflammation, or neuroinflammation, is an essential driving factor of many neurodegenerative diseases [17,18]. Abnormal activation of microglia in a patient’s brain is associated with this disease pathology [19]. Proinflammatory cytokines, chemokines, or reactive oxygen species released by activated microglia potentiate the interaction between neuroinflammation and α-Synuclein dysfunction, which drives the progression of Parkinson’s disease (PD) [20]. In this context, immunized neural organoids serve as an invaluable tool in the detailed exploration and understanding of neurological diseases [21]. This comprehensive review centers on the crucial role of microglia within immunized brain organoids, delving into their impact on understanding neuroinflammation in neurological diseases. It covers technological advancements, potential applications in disease research, and the current challenges and limitations in the field.

## 2. Interaction between Microglia and Neurons

### 2.1. Patterns of Interaction between Microglia and Neurons

In the nervous system, interactions between microglia and neurons could be facilitated through a variety of mechanisms. These include the release of soluble factors, direct physical contacts between cells, or through indirect interactions that involve intermediary cell types [22].

Soluble messengers are crucial for bidirectional communication between microglia and neurons. This interaction is facilitated through the release of substances that act at a distance from their origin, enabling microglia and neurons to communicate effectively. For instance, microglia-derived brain-derived neurotrophic factors (BDNF) not only play an essential role in transmitting neuropathic pain signals to neurons, but also lead to a shift in the anion reversal potential of spinal lamina I neurons [23]. Additionally, microglia-released interleukins, such as IL-10 and IL-1β, influence neuronal development and activity [24,25]. In the context of neuronal influence on microglia, recent studies have shown that microglia can detect neuronal ATP and respond by producing adenosine, which then inhibits neuronal activity [26]. During traumatic brain injury, macroglia can form a potential barrier between healthy and injured tissue in an ATP-dependent manner, highlighting the dynamic nature of neuroglial interactions in response to injury [27].

The direct microglia-neuron interaction mostly takes place on the microglia-synapse interface [28]. Several molecules have been identified in microglia-neuron contacts at synapses, including Fractalkine (CX3CL1)/fractalkine receptor (CX3CR1) [29], CD200/CD200R [30], and the complement system [31]. The CX3CL1-CX3CR1 interaction is crucial for synaptic pruning by microglia [29], as highlighted by the impaired functional maturation of synapses in CX3CR1-deficient mice [32]. Microglia contribute to neurogenesis through crosstalk with neural precursors in the developing cortex [33]. They also have the capacity to direct neuronal migration and differentiation [34]. Synaptic pruning requires microglia during brain development, necessary for refining synaptic circuits [29]. CX3CR1, a vital chemokine receptor essential for microglial function, is instrumental in synaptic pruning, a key process in maintaining neural network health and adaptability. Deficiencies in CX3CR1 lead to compromised synaptic pruning, resulting in diminished synaptic transmission and decreased brain connectivity. This impairment can subsequently trigger the onset of various neuropsychiatric disorders, highlighting the receptor’s critical role in brain function and mental health [35]. CD200 acts as a negative regulator for microglia. When CD200 is deficient, it can lead to an excessive increase in both the number and activity of microglia, consequently increasing the occurrence of experimental autoimmune encephalomyelitis [36].

Intermediate cells like astrocytes simultaneously receive and transmit signals from both microglia and neuros [37]. Continuous exchanges of metabolites occur between neuros and astrocytes [38]. Moreover, whereas astrocytes and microglia collaborated with glutamatergic neurons, constitute the “quad-partite synapse.” This structure is crucial for neural activity and serves as a pivotal component in neuro-immune communication, highlighting the intricate and integrated nature of neuronal and immune system interactions in the brain [39].

### 2.2. The Significance of Microglia and Neuron Interactions

The role of interactions between microglia and neurons may be context-dependent. Microglia perform surveillance functions through constantly monitoring their environment with highly motile processes under homeostatic conditions [40]. Acute and rapid activation of microglia is generally considered neuroprotective [41], whereas persistent and prolonged microglial activation may be neurotoxic, ultimately leading to neuronal degeneration [42]. Emerging data suggest that neuroimmune dysfunction in microglia-synapse interactions contributes to neurodegenerative diseases. In the early stages of Alzheimer’s disease (AD), complement and microglia are implicated in synapse loss, characterized by increased C1q in synapses and subsequent microglial phagocytosis before the deposition of pathological plaques. Blocking C1q or the complement receptors on microglia protect against early synapse loss, indicating that targeting microglia-synapse interactions could be a potential therapeutic approach in AD [31]. In PD, the accumulation of α-synuclein in synapses suggests a similar engulfment mechanism as seen in AD, although the role of complement in synapse loss in PD remains largely unknown [43]. Reactive astrocytes, induced by activated microglia, contribute to direct neurotoxicity in neurodegenerative diseases [44], and regulating microglia-astrocyte crosstalk demonstrates a potent neuroprotective effect in PD [45]. Enhancing the CX3CL1-CX3CR1 signaling pathway also plays a neuroprotective role in a PD mouse model [46].

## 3. From iPSC-Derived Microglia to Immunized 3D Neural Organoids

Due to their unique anatomical location, human microglia are almost exclusively explored through in vitro research methods. However, the availability of primary microglia for such studies presents substantial challenges. Various microglial models have been utilized to elucidate the roles of microglia in neurological diseases and to develop potential therapeutic development. These models encompass postmortem microglia [47], microglia derived from iPSCs [48], two-dimensional (2D) co-cultures of microglia and neurons [49], 3D organoids containing microglia [50], and even more intricate organoid systems [51]. The emergence of iPSC technology marked a significant breakthrough, paving the way for more comprehensive and effective studies of human microglial cells.

Muffat et al. were the first to describe microglia derived from iPSCs. A fully defined serum-free medium that mimics the central nervous system environment was developed for the derivation of microglia-like cells in human iPSCs. The medium contains CSF1 and IL-34, which are critical for the survival and maturation of microglia [48]. Subsequent studies published similar strategies to generate iPSC-derived microglia (iMGs), while each protocol is different in specific details [52,53]. Embryoid bodies (EBs) were employed in Muffat’s study for the generation of microglia in an early step [48], whereas Douvaras et al. used monolayer cultures instead of EB [52]. Although the two protocols are comparable in efficiency, the latter requires less iPSCs. Another protocol improved the differentiation of iPSCs into microglia by simplifying the initial medium and adding glial cell-derived cytokines [53].

Transcriptomic analysis has verified that iMGs closely resemble primary microglia found in both human fetal and adult tissues. iMGs express key microglial markers, including TMEM119, P2RY12, and CX3CR1. iMGs also demonstrate the ability to replicate essential microglial functions. These include migrating to injury sites in response to damage within 3D culture systems, exhibiting phagocytic activity towards foreign substances, and releasing cytokines when stimulated by Lipopolysaccharides (LPS) or Interferon-gamma (IFN-γ). The findings underscore the potential of induced microglial cells as an effective model for studying the behavior of microglia in the context of neural disease pathogenesis [48,52,53].

Although genome sequencing has identified numerous genes associated with neurological disorders, understanding how these genes contribute to disease development remains a challenging area of study. Exposing iMGs to brain substrates like myelin debris, synthetic amyloid-β(Aβ) fibrils, synaptosomes, and apoptotic neurons has helped generate and identify diverse transcriptional states in microglia, including a neurodegenerative disease-associated microglial (DAM) state [54]. The overexpression or mutation of key genes linked to Parkinson’s disease progression, such as SCAN and LRRK2, has been identified. However, the difficulty in obtaining sufficient human tissue hindered further studies of these genes in microglia [55,56]. iPSCs have been used to create microglia-like cells from familial PD patients with a triplication or A53T mutation of the SCAN gene. These iMGs exhibited reduced phagocytosis when exposed to high levels of α-synuclein. Different mechanisms have been observed in the microglial endocytosis of fibrillar and monomeric α-synuclein: fibrillar α-synuclein is taken up through actin-dependent pathways, while monomeric α-synuclein is absorbed via actin-independent pathways [57]. This provides valuable insights into how microglia respond to excess α-synuclein. In a study focused on AD, researchers utilized an integrated, automated culturing platform comprising neurons, microglia, and astrocytes derived from iPSCs. That study confirmed that human iMGs exhibit neuroprotective effects, particularly by internalizing and compacting Aβ. However, it was also found that in instances of neuroinflammation, microglia lose this neuroprotective ability, which in turn exacerbates the progression of AD [58].

Given the limitations of 2D cell cultures in accurately mirroring the intricate structure and unique characteristics of the human brain, 3D-cultured organoids emerged as a superior alternative, offering a more complex and representative microenvironment for research purposes [50]. The unintended activation of microglia during in vitro studies is a critical concern that requires careful management [59]. When microglia are isolated from their natural in vivo environment and cultured in vitro, they experience a phenomenon known as ‘culture shock’. This transition leads to the increased expression of certain disease-related genes, including APOE, LYZ2, and SPP1, within the cultured microglia, as opposed to the levels observed in primary microglia [60]. However, when comparing microglia cultured under traditional monolayer conditions to those integrated into organoids, the latter tend to maintain a more rudimentary and ramified-like form [21]. This suggests that microglia in organoids more closely mimic their natural, resting state found in vivo, highlighting the importance of the culture environment in maintaining physiological relevance. Furthermore, when these microglia-integrated immunized human brain organoids are transplanted into the cranial cavity of mice, the microglia demonstrate the capacity to respond not only to localized injuries but also to systemic inflammation [21]. This highlights their potential for studying microglial behavior and responses in a more physiologically relevant context. However, whether microglia in organoids cultured in vitro truly reflect the state of microglia in the human central nervous system still requires further study.

## 4. Strategies for Immunizing Neural Organoids

### 4.1. Innate Development of Microglia in Neural Organoids

Neural organoid immunization primarily focuses on the development and differentiation of immune cells, particularly microglia-like cells, to explore and manipulate the immune responses within the organoid. Although the protocols generally used to prepare neural organoids do not yield immune cells, there have been reports of mesodermal-derived progenitors emerging in neural organoids [61], which can even differentiate into microglia. These innately developed microglia exhibit typical microglial morphology, express relevant cell surface markers, and display phagocytic capability [62]. In research involving 2D ocular organoids, cells resembling microglia and positive for PAX6 were identified. However, these cells demonstrated limited phagocytic capabilities [63]. The protocol for innately generating microglia in organoids often varies among different studies. However, there is currently no direct evidence to suggest that these modifications result in the production of Hematopoietic Progenitor Cells (HPCs) or provide the necessary conditions for the differentiation of HPCs into microglia. Consequently, researches into using spontaneously generated microglia within organoids for immunotherapy is still in its infancy, with methodologies across different studies lacking standardization. A recent method that has emerged involves ectopic expression of PU.1 in pluripotent stem cells, mixing them with standard pluripotent stem cells, and subsequently forming neural organoids. PU.1 serves as a crucial transcription factor for myeloid differentiation, providing a robust induction signal for stem cells to differentiate into microglia [64]. This method presents a novel strategy for the consistent generation of microglia within neural-like organs. While it shows promise, it still requires additional research and validation for further development.

### 4.2. Integration of iPSC-Derived Microglia into Neural Organoids

The primary approach for creating immunized brain organoids involves differentiating iPSCs into microglia and neural organoids separately, which are then integrated. This key step of producing HPCs is not present in the differentiation process of neural organoids [65]. The separate differentiation followed by fusion is an effective method for establishing microglia-containing immunized organoids [66]. The integration with neural organoids can involve either microglia progenitor cells or more mature microglia. The fusion of microglia progenitors with neural organoids may provide more opportunities for interaction with the nervous system during the development of microglia, thereby better simulating their developmental process within the neural system. On the other hand, terminally differentiated microglia might offer better control over the ratio of microglia to neural organoids. These specific methods and their analyses have been described in other excellent reviews [67]. The initial step in generating microglia from iPSCs involves inducing their differentiation into HPCs, the precursors of microglia. Several protocols are available for this differentiation process, with some requiring hypoxic conditions [13] and others utilizing EBs [68]. Both approaches can effectively produce high-quality HPCs, which can subsequently be induced to form microglia. These methods have also been reported to integrate successfully with brain organoids [21,66]. Among these methods, the EB-based strategy is particularly recognized for its ease of standardization, which is also employed by commercial kits. For EB formation, iPSCs need be digested into single cells, and then resuspended in EB medium at a concentration of approximately 100,000 cells per milliliter. A 100-microliter cell suspension should be seeded into a low-attachment round-bottomed 96-well plate, resulting in a seeding density of 10,000 cells per well. Cell aggregates are formed by spinning down the plate. Subsequently, the plate should be gently transferred to the incubator for further cultivation. Besides the basic culture medium, EB medium typically includes supplements such as Rock-inhibitor (Y-27632), BMP-4, SCF, and VEGF to enhance survival and induce hematopoietic differentiation. This EB formulation is designed to mimic the natural environment of microglia production and is sometimes referred to as yolk sac EB in the literature [66]. EBs will attach to the surface of culture plates, and the medium will be supplemented with M-CSF, IL-34, and TGF-β. After continuing the culture for about 2–3 weeks, precursors of microglia will appear in the supernatant. This process can continue for several weeks, and microglia precursors can be collected from the supernatant by changing the culture medium [13]. These collected precursor cells can be co-cultured with organoids, allowing the precursors to further differentiate and mature into microglia in the neural microenvironment.

iPSC-derived brain organoids start with neural induction. In this step, iPSCs are dissociated into clumps and then allowed to form 3D structures in low-attachment cell culture plates. The neural induction stage requires about 9 days, during which the culture medium, based on iPSC medium, is supplemented with SMAD inhibitors, SB-431542 and dorsomorphin, as well as Y-27632 to promote cell survival. After the neural induction phase, starting from day 9, the medium is switched to a neural differentiation medium composed of Neurobasal media, supplemented with B27, N2, FGF, and EGF, among others. Neural differentiation will continue until the 25th day. From day 25 onward, growth factors like FGF and EGF should be omitted from the culture medium. Subsequently, the medium needs to be refreshed every four days to sustain the culture’s health and progression. The length of time for maintaining the culture beyond this point will vary, tailored to the specific research objectives and the neural functions of interest [50,69]. The process of differentiating iPSCs into brain organoids generally spans approximately 35 to 45 days, and the development of advanced neuronal function within these organoids may require additional time. Determining the optimal moment to incorporate microglia progenitors into brain organoids is a crucial consideration. The appropriate timing for this integration can vary, largely depending on the specific objectives of the scientific studies. Research suggests that, to maximize microglia viability, the best window for integrating microglia progenitors might be during the later phases of neural differentiation, typically around days 35 to 42 [21]. Following the integration of microglia progenitors with organoids, another critical challenge is the selection of a suitable culture medium that supports the coexistence and growth of both microglia and neurons within the immunized brain organoids. Experimental evidence suggests that the neural microenvironment offered by brain organoids alone is inadequate for microglia development. Therefore, to ensure the survival of microglia incorporated into the organoids, it is essential to supplement the standard brain organoid culture medium with additional cytokines, such as M-CSF, IL-34, and TGF-β [21].

## 5. Regionally Specific Immunized Organoids

Immune cells residing in various tissues often exhibit unique biological characteristics. However, limited research has explored whether microglia in different regions of the central nervous system possess distinct traits. This knowledge gap may be attributed to the challenges associated with obtaining samples from the central nervous system.

Several studies have initiated investigations in this area. A research differentiating iPSCs into two distinct regions of the cerebral cortex: the dorsal and ventral forebrain, revealed that the integration of microglia into different brain regions led to diverse outcomes, influencing their migration capabilities and responses. When exposed to amyloid-beta 42 (Aβ42) stimulation, microglia integrated with the dorsal forebrain demonstrated an increased secretion of anti-inflammatory factors. In contrast, those integrated with the ventral forebrain showed heightened production of pro-inflammatory cytokines [70]. Although further research is needed to enhance the stability and accuracy of differentiation in specific brain regions, this study offers new insights and approaches for exploring microglial functions across different anatomical areas within the central nervous system.

The midbrain is a susceptible organ in various neurological diseases, with PD being the most prominent among them. A critical feature of PD is the loss of dopaminergic neurons in the substantia nigra of the midbrain [71]. Abnormalities in microglial function are frequently linked to the onset of the disease [72]. Animal experiments have validated that microglia in the midbrain exhibit distinct characteristics. For instance, distinct from the cortex, hippocampus, and striatum, the midbrain is noted for containing two unique populations of microglia: one group exhibits high levels of MHC-II expression, while the other prominently expresses TLR4. These microglia demonstrate an elevated responsiveness to inflammatory signals. However, within the midbrain environment, there is a tendency to observe an immunosuppressive response, notably marked by a decrease in MHC-II expression and an increase in anti-inflammatory mediators such as IL-10 and TGF-β [73]. The underlying mechanisms of this phenomenon represent an intriguing field of study. It also indicates that research into the relationship between microglia and the human midbrain is still relatively limited, especially within the context of PD. The integration of microglia within midbrain organoids has the potential to fill this gap. Initial studies have found that microglia in midbrain organoids can influence the expression of the genes associated with synaptic remodeling and enhance neuronal excitability [74]. Subsequent research in this area will likely aid in explaining the mechanisms underlying midbrain-related diseases and exploring potential therapeutic approaches.

The Blood-Brain Barrier (BBB) serves as a vital interface between the central nervous system and the circulatory system, playing a key role in maintaining brain homeostasis and protecting neural tissue from harmful substances and pathogens. Adjacent to this barrier lies a unique set of non-parenchymal, tissue-resident macrophages. These cells are located within the perivascular spaces and are termed border-associated macrophages (BAMs) [75]. BAMs are distinct from the parenchymal microglia and are crucial for maintaining brain health and immune surveillance [76]. Considering that conventional brain organoids lack a functional vascular network, the creation and amalgamation of vascularized brain organoids to simulate immune functions offer a promising methodology for the exploration of BAMs [77]. Research that separately cultivates vessel and brain organoids before combining them to form vascularized brain organoids has shown that macrophages can form around the vasculature. In these integrated organoids, BAMs not only exhibit specific macrophage markers and react to immune stimuli, such as LPS, but they also demonstrate the ability to engulf synapses [78]. This significant finding opens new pathways for studying how the brain’s vascular and immune systems interact, enhancing our understanding of their roles in brain health and disease. However, questions remain regarding the authenticity and functional similarity of the generated BAMs to their in vivo counterparts, their distinctions from microglia, and the necessity of incorporating iPSC-derived macroglia for a more accurate representation of the brain’s immune environment.

## 6. Immunized Organoids in Neurodegenerative Diseases

### 6.1. Alzheimer’s Disease

The study of AD pathologies in microglia-containing organoids has been advanced through a variety of strategies. These include the use of isogenic organoids, patient-derived organoids, and organoids induced with Aβ treatment. This diverse methodology enriches our comprehension of the role of microglia in AD pathogenesis (Figure 1). Subsequent research has expanded on these foundational models, offering deeper insights into the genetic and pathological aspects of AD.

APOE4, a variant of the apolipoprotein E gene, is widely recognized as a significant genetic risk factor for AD. It plays a critical role in lipid transport and injury repair in the brain. Individuals carrying the APOE4 allele have a higher risk of developing AD, and it is associated with an earlier onset of the disease [79]. Neurons carrying the APOE4 allele showed a heightened number of synapses and an increased secretion of Aβ42 compared to their APOE3 counterparts. Furthermore, APOE4 microglia-like cells demonstrated distinctive morphological changes that were associated with a reduced capability for Aβ phagocytosis. These findings highlight the multifaceted influence of APOE4 across different brain cell types, with a particular emphasis on the altered function of microglia in Aβ clearance, underscoring their potential key role in the development of AD pathologies [80]. While GWAS has identified numerous AD-associated genes [81], there is limited direct evidence to conclusively confirm their contributions to the onset mechanisms of AD.

Utilizing CRISPR interference (CRISPRi) and CRISPR droplet sequencing techniques, specific AD-related genes, including TREM2, CD33, and SORL1, were selectively knocked out in the immune cells of Aβ-treated immunized neural organoids. While this genetic intervention did not significantly affect the microglial population size, it profoundly altered their functional attributes as microglia. The absence of TREM2 was found to potentially affect the clustering of microglia around Aβ peptides. Moreover, SORL1 was observed to play a role in maintaining a normal cholesterol turnover rate by promoting the expression of CYP461 [64]. TREM2 and SORL1 are both genes associated with APOE in microglia [82,83]. The role of APOE in AD has also been confirmed in a wider range of immunized organoids. Immunized organoids derived from the APOE4 genotype exhibit decreased autophagy levels, which could potentially contribute to the development of cerebral amyloid pathology [84]. Down syndrome (DS) is identified as one of the primary risk factors for AD. While not all individuals with DS are affected, many tend to develop AD as they age [85]. Based on organoids immunized with microglia derived from iPSCs, it has been observed that microglia from DS patients show an enhanced or excessively strong capability for synaptic pruning. Moreover, upon exposure to pathological tau proteins, DS microglia demonstrate an increased activation of type 1 interferon signaling pathways [86]. Targeting the interferon alpha/beta receptor (IFNARs) emerges as a promising approach to potentially improve the functionality of DS microglia and offer a therapeutic avenue for AD. Via immunized neural organoids, it has been discovered that microglia-derived cardiolipin enhances the endocytosis of Aβ by astrocytes and microglia themselves. The decrease in cardiolipin levels as one ages could potentially contribute to the onset of AD [87]. A decline in cardiolipin levels with aging may contribute to the pathogenesis of AD.

### 6.2. Parkinson’s Disease and Atrial Septal Defect

Leucine-rich repeat kinase 2 (LRRK2) is a gene significantly associated with PD. Mutations in this gene, particularly the G2019S mutation in certain populations, represent one of the most common genetic causes of PD [88]. Compared to normal counterparts, astrocytes with the G2019S mutation exhibit reduced levels of MMP2 and TGFB1 [89]. Both MMP2 and MMP3 play a key role in degrading α-synuclein [90], whose abnormal accumulation is believed to be a significant contributing factor to PD [91]. TGF-β, on the other hand, is an important cytokine that negatively regulates inflammation mediated by microglia [92]. A decrease in TGF-β could potentially lead to excessive activation of microglia. Research has shown that microglia originating from iPSCs with LRRK2 mutations demonstrate similarities in gene expression and functionality to microglia found in patients with PD [93]. It has also been revealed that mutations in the LRRK2 gene within microglia can affect their motility and the expression of adhesion molecules [94]. However, there has not yet been a comprehensive study using neural organoids to explore in depth how LRRK2 mutations impact neural cells and their surrounding microenvironment in the context of PD.

Microglia in immunized brain organoids derived from Atrial Septal Defect (ASD) patients exhibit more primed morphology, fewer resting states after being transplanted into immune-deficient mice, along with larger soma and increased primary process thickness. Both microglia and the neural environment may contribute to the excessive inflammation. To confirm this, researchers introduced normal microglia into neural organoids derived from ASD patients and observed that the ASD neural environment induced similar morphological changes in the normal microglia, mirroring those observed in ASD-derived microglia [21]. This study not only reveals that alterations in the brain environment due to disease significantly contribute to the hyperactivation of ASD-associated microglia, but it also offers fresh perspectives on the interactions between neural cells, the neural microenvironment, and microglia in the development of neurological disorders.

### 6.3. Multiple Sclerosis

From an immunological perspective, multiple sclerosis (MS) is recognized as a chronic autoimmune disorder where the immune system erroneously targets and damages the myelin sheath, the protective covering of neurons in the central nervous system, leading to disrupted neural communication and varied neurological symptoms. Although immune responses mediated by T cells and B cells are crucial in the development of MS [95], microglia could also play a pivotal role in the disease’s onset [96]. Microglia might present self-antigens to T cells, triggering an immune response and prompting B cells to create antibodies against these antigens. Neurons opsonized by self-antibodies become ideal targets for microglial phagocytosis, leading to their increased susceptibility to immune-mediated destruction. Genetic mapping has revealed numerous MS susceptibility genes linked to microglia, highlighting their critical function in the pathogenesis of MS [97]. However, current research, mainly grounded in autopsies and animal studies, falls short of accurately representing the true involvement of human microglia in MS [98]. The pathogenic factors of MS involve a complex interplay between genetic and environmental triggers, along with intricate cellular alterations in both neural and immune cells. Organoids prepared from iPSCs derived from healthy individuals showed fewer cell proliferation markers and stem cells compared to those from healthy donors. Additionally, organoids from MS patients exhibited a higher number of mature neurons and a reduced count of oligodendrocytes. These findings indicate abnormalities in the cells derived from neural stem cells, which may impact the progression of the disease [99]. Research employing brain organoids to study MS remains in its early stages. Studies incorporating immunized organoids to explore the roles of cells derived from hematopoietic stem cells, such as lymphocytes, microglia, or infiltrating macrophages, in MS’s cellular and molecular mechanisms, are lacking. Unlike AD and PD, the potential abnormalities in the microglia associated with MS may be more complex [16], potentially involving interactions with T cells and antibodies. This intricacy poses substantial challenges to developing immunized organoids that accurately simulate the immune microenvironment specific to MS.

### 6.4. Infection Diseases

Based on immunized organoids, it was found that viral infections can cause neurocircuit integrity damage by activating microglia-mediated synapse elimination, thereby manifesting a phenotype similar to neurodegenerative disorders [100]. Although research on virus infection in neural organoids has demonstrated direct damage to neural cells, it is important to note that the absence of immune cells in neural organoids limits the understanding of the immune response and neuroinflammation triggered by infection [101]. When the Zika virus infects immunized organoids, it leads to the activation of microglia, resulting in increased expression levels of pro-inflammatory cytokines such as IL-6, IL-1β, and TNF-α. Additionally, the expression of type 1 interferon receptors within microglia also rises, indicating an enhanced ability to respond to inflammatory signals. Furthermore, the phagocytic ability of microglia within immunized organoids significantly improves [66]. These findings strongly suggest that microglia within immunized organoids following Zika infection may possess an augmented, or possibly even an excessively increased, synaptic pruning capability. Additionally, another study reported similar findings, indicating that SARS-CoV-2 infection can undermine neural circuit integrity through the overactivation of microglia, leading to excessive synaptic pruning [100].

## 7. Challenges and Limitations of Immunized Brain Organoids

The integration of microglia into brain organoids has been accomplished effectively, and the resulting immunized organoids have exhibited notable potential in a wide range of disease contexts. Nevertheless, continuous challenges remain that hinder the progression and broader application in the field of neurology and immunology.

### 7.1. The M1/M2 Paradigm in Brain Organoids with Integrated Microglia

The distinction between pro-inflammatory M1 cells and the more reparative M2 cells marks a standard framework for classifying peripheral macrophages. Similarly, this segregation has been observed in microglia residing in the central nervous system, showcasing comparable phenotypic divisions [102]. Notably, microglia integrated into neural organoids are capable of displaying M1 traits, such as the secretion of IFN-γ and exhibiting phagocytic activities [13]. Moreover, microglia derived from iPSCs can replicate various M2 traits and have been implemented in disease research projects. For instance, iMGs with M2-like features may exhibit enhanced immunosuppressive effects, notably in inhibiting effector T cells and facilitating the induction of regulatory T cells [103]. However, the investigation into M2-typed microglia within brain organoids, particularly with respect to neurological disorders, remains underexplored. Investigating the classical and alternative activation of microglia in immunized brain organoids provides a valuable opportunity not just to ascertain whether the M1 and M2 designations correspond to a continuum of functionalities or separate categories, but also to identify potential therapeutic targets for neuroinflammatory diseases.

### 7.2. Heterogeneity of iPSC-Derived Microglia

Advancements in high-resolution technologies, such as single-cell RNA sequencing (scRNA-seq), have led to the discovery of a broader spectrum of microglia subgroups [104]. These subgroups are challenging the traditional binary classifications such as M1 or M2, or simplistic distinctions like pro-inflammatory versus anti-inflammatory, or universally good versus bad microglia. Instead, the characterization of these subgroups is being reshaped, influenced by a comprehensive analysis of factors including gene expression profiles, morphological features, and epigenetic markers [105]. Once microglia are removed from their native in vivo environment to an in vitro setting, their heterogeneity significantly reduces. This reduction also extends to microglia derived from iPSCs, which maintain lower levels of heterogeneity even when incorporated into brain organoids. However, this heterogeneity notably increases when immunized brain organoids are transplanted back into an animal model, highlighting the critical role of in vivo factors in preserving microglia heterogeneity [50]. Current research on immunized organoids implanted into animal models is limited, but these findings suggest that maintaining an in vivo environment may be crucial for preserving microglia heterogeneity. This is particularly relevant in research on neuroinflammatory diseases using immunized organoids to study the role of microglia. The in vivo context might significantly enhance the accuracy and relevance of such studies, highlighting the complex dynamics of microglia behavior and their contributions to disease pathogenesis.

### 7.3. Immune Cell Diversity in Brain Organoids

Another challenge in the development of immunized organoids is that the research primarily focuses on the incorporation of microglia. Although microglia represent the central immune cells in the central nervous system, recent studies have gradually unveiled the presence of additional types of immune cells within this system [106], most notably T cells [107]. In AD animal models, Aβ-specific Th1 and Th17 cells have been observed to exacerbate memory impairment and systemic inflammation, alongside an increase in microglia activation [108]. In contrast, Aβ-specific regulatory T cells have been demonstrated to inhibit the activation of microglia, thereby mitigating the accumulation of amyloid plaques [109]. These discoveries highlight the complex interplay between different immune cells within the central nervous system and underscores the potential for broader immunological approaches in brain organoid research. However, a considerable portion of this research relies on animal models, particularly studies involving mouse T cells and mouse microglia. The question remains whether the mechanisms observed in mice parallel those in humans, specifically regarding the interaction between human T cells and microglia within the human nervous system. This gap highlights the necessity for additional research and validation. Furthermore, the specific functions of T cells, particularly Th cells, may require the participation of microglia to effectively influence neurological cells or conditions. Therefore, immunized brain organoids could serve as a valuable platform for advancing research in this area. Nevertheless, the representation of immune cells other than microglia in these organoids remains strikingly limited at present. Expanding the diversity of immune cells within brain organoids would not only enhance our understanding of the regulatory mechanisms between adaptive and innate immune responses within the neural microenvironment, but also promote the study of the complex interactions between the immune and nervous systems. This approach could significantly contribute to elucidating the roles and contributions of neuroinflammation in maintaining neural system stability and in the pathogenesis of neurological diseases.

### 7.4. Immaturity of iPSC-Derived Microglia

The maturation of microglia is dependent on signals provided by the central nervous system’s microenvironment. These signals could originate from neurons or other cells; they might encompass cytokines or derive from direct intercellular contact. Microglia cells induced from iPSCs might face challenges related to insufficient maturation. While iMGs may display certain surface markers that are characteristic of mature microglia, current studies remain inadequate to affirm their functional maturity in comparison to the standards of fully developed microglia [13]. Reports indicate that specific microglial functions, such as the capability for phagocytosis, are diminished in iMGs compared to macrophages, suggesting a less mature state from a functional standpoint. Microglia integrated into brain organoids, in contrast to those cultured in two dimensions, exhibit an increased number of cellular processes, indicating a closer resemblance to the mature microglia found within the stable in vivo environment [21]. Furthermore, the in vivo environment provided by immunodeficient mice might also offer conducive conditions for promoting the maturation of human iMGs [110]. Microglia within immunized organoids transplanted into immunodeficient mice demonstrate a greater degree of developmental maturity, showcasing more extensively the branched, or ramified, state typical of mature microglia under homeostatic conditions [21]. Nevertheless, the more sophisticated functions of mature microglia or macrophages, such as the ability to present antigens and activate T cells, currently lack comprehensive investigation within the context of immunized organoids.

### 7.5. Ethical Concerns in Relation to Immunized Brain Organoids

Immunized brain organoids represent a major breakthrough in biomedical research, providing unique opportunities to explore neurological diseases, brain development, and the interactions within the immune system in a detailed, three-dimensional setting. On one hand, iMGs offer a novel source for microglia research, circumventing the ethical dilemmas tied to acquiring microglia from post-mortem specimens or aborted fetuses. Nonetheless, the utilization of immunized brain organoids introduces several ethical challenges. The creation of organoids largely depends on human iPSCs, which originate from adult tissue. The source of these cells can spark ethical controversies. Issues such as informed consent, privacy protection, and the risks of commercial misuse are of paramount importance. As microglia contribute to making brain organoids more advanced in terms of neural signaling and functionalities, there emerges a possibility that these organoids could exhibit aspects of human consciousness. Even though there is no current evidence to support the presence of human consciousness in these structures, it is crucial to proactively discuss the ethical boundaries concerning the manipulation of such human-cell-derived tissues [111]. Furthermore, a significant ethical dilemma emerges when immunized brain organoids are transplanted into animals, resulting in part-human chimeras. The appropriate implications of treating these creatures necessitate thoughtful and proactive discussion [112].

## 8. Concluding Remarks

The difficulties associated with obtaining primary human microglia, particularly from individuals across different ages and from patients with diverse diseases, have impeded progress in neuroimmune science. iMGs have become an important alternative, mitigating some of the sourcing challenges faced by researchers in this domain. Current research on immunized brain organoids primarily focuses on the study of neurological diseases. However, their application should be further expanded. For instance, the potential link between microglial abnormalities and the changes observed in the aged brain represents an intriguing and significant area of study. Should immunized organoids yield breakthroughs in understanding age-related alterations within the nervous system, and subsequently lead to the development of strategies to decelerate brain aging in humans, it would signify a significant advancement. Immunized brain organoids serve as an innovative platform for neuroimmunological research, yet this technology is still in a nascent phase, with experimental methods lacking uniformity, standardization, and quality control. There is an urgent need for additional researchers to refine this technology and extend its applications to a broader range of research fields, thereby realizing its full potential in diverse scientific studies.

## Figures and Tables

**Figure 1 ijms-25-03148-f001:**
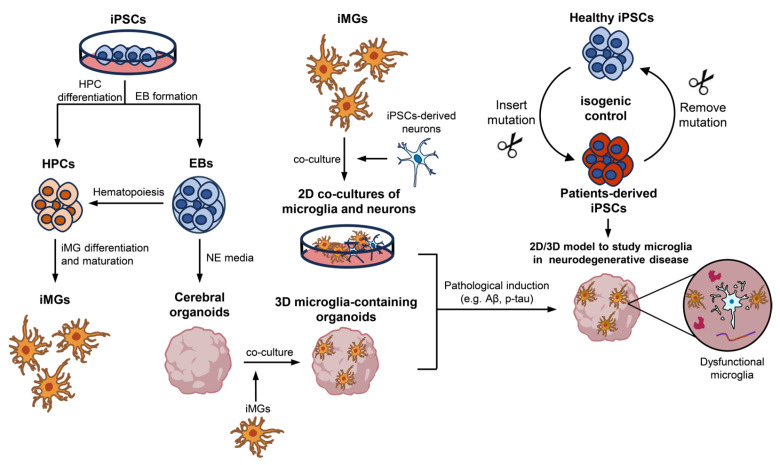
Induced pluripotent stem cell (iPSC)-derived microglia immunized brain organoids in neurodegenerative disease studies. iPSC-derived microglia (iMGs) were generated by first forming embryoid bodies (EBs) for Hematopoietic Progenitor Cells (HPCs) generation from iPSCs, and then differentiating into iMGs. Meanwhile, iPSCs were induced into neuroepithelium differentiation, with self-assembly into cerebral organoids. Neuron-microglia interactions at 2D–3D levels were achieved by co-culturing iMGs with iPSC-derived neurons or cerebral organoids. To model neurodegenerative diseases using 2D co-cultures or 3D organoid cultures, mutations can be introduced into healthy control iPSCs or corrected in patient-derived iPSCs using gene-editing techniques, while maintaining the original genetic background of the iPSC line. Chemical induction methods, such as employing Aβ and p-tau to simulate Alzheimer’s disease pathology, can be utilized to induce specific pathological conditions in both 2D co-cultures and 3D organoid models.

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
