# Peer review of "Integration of iPSC-Derived Microglia into Brain Organoids for Neurological Research"

_ijms, 2024, doi:10.3390/ijms25063148_

Round 1

Reviewer 1 Report

Comments and Suggestions for Authors

Asif Mrza et al. offer an overview of recent advancements aimed at bridging a crucial gap through the integration of microglia into iPSC-derived neural organoids, creating an immunized microenvironment that better mirrors the human central nervous system. Despite its significance in the field, this review falls short in articulating a distinct and conclusive outcome.

The major concerns identified are as follows:

Title Modification:

The title requires modification for clarity and precision. I suggest revising it to: "Integration of Microglia into iPSC-Derived Neural Organoids for Neurological Research."

Grammar, Syntax, and Readability:

The manuscript suffers from numerous grammar inconsistencies and lacks clarity in syntax. While I attempted to address some issues, the pervasive nature of these errors necessitates a comprehensive review by a language expert.

Inclusion of Key Diseases:

The manuscript inadequately covers diseases such as Parkinson's (a-synuclein), some forms of Alzheimer's (APOE4), and autism spectrum, yet it overlooks crucial elements such as multiple sclerosis (MS) and aging. Addressing these omissions is essential for a more comprehensive study.

Microglia Origins:

The origin of microglia is poorly described. I recommend consulting the review at https://pubmed.ncbi.nlm.nih.gov/36975541/ to enhance the clarity and depth of this section.

Fusion of Microglia Progenitors with Neural Organoids:

The manuscript outlines the process of fusing microglia progenitors with neural organoids, but it lacks a thorough explanation of the implications and outcomes of this procedure. A more detailed elucidation is needed to enhance the understanding of the readers.

Critical Analysis and Perspectives:

The remarks section lacks depth and fails to provide adequate perspectives. A comprehensive discussion of the drawbacks, coupled with potential future directions, is necessary for a more balanced and informative manuscript.

Blood-Brain Barrier (BBB) Section:

Paragraph line 66 regarding the Blood-Brain Barrier (BBB) is insufficient. I suggest expanding on this topic by including relevant information from https://www.mdpi.com/1467-3045/45/5/272, particularly focusing on the Border Associated Macrophages (BAMs).

Figure 1:

The quality and design of Figure 1 are subpar, and the pipeline lacks clarity with undefined outcomes. A revision of the figure, with a clearer representation of the process and outcomes, is essential for effective communication.

I appreciate the effort authors have invested in this research, and I believe that addressing these concerns could significantly improve the manuscript's quality before considering submission to another journal.

Comments on the Quality of English Language

The manuscript exhibits a notable number of grammar errors, which, collectively, significantly impede the clarity and coherence of the content. A thorough revision by a language expert is imperative to rectify these issues.

Reviewer 2 Report

Comments and Suggestions for Authors

The authors compiled a review on organoids for neurological research. This is a very important field that could answer a lot of the questions that cannot easily be answered/tested in animal models of disease.

However, the review could use a few more additional pointers that will give a more rounded view of the field for readers.

1.         Microglia are easily activated in ex-vivo conditions. The authors need to discuss further on the activation state of microglia in organoids. 

Ref: Microglia-containing human brain organoids for the study of brain development and pathology. Wendiao Zhang et al.

2.         T cells are a major component of microglia activity/activation state. The authors need to add a section on the limitations/feasibility of adding T cells to such organoids.  

For example, the authors mention at line (167) that microglia loose the ability to internalize Aβ during neuroinflammation. Disease specific T effector cells (Teff) have been shown to exacerbate amyloid pathology, Jatin Machhi et al. CD4+ effector T cells accelerate Alzheimer’s disease in mice. While disease specific T regulatory cells (Treg) have been shown to drive a non-reactive phenotype and increase plaque clearance.  Amyloid-β specific regulatory T cells attenuate Alzheimer’s disease pathobiology in APP/PS1 mice Pravin Yeapuri et al.

It would be of great clinical value if an organoid with microglia and Aβ plaque can be infused with Aβ-specific Treg or Teff…,  and organoids infused with Treg facilitates increased microglial clarence of Aβ. This can be replicated across many neurodegenerative diseases such as Parkinson’s disease and ⍺-syn.

Comments on the Quality of English Language

English is fine, no major edits needed.

Reviewer 3 Report

Comments and Suggestions for Authors

The review article entitled 'iPSC-Derived Microglia Immunize Organoids for Neurological Research' by Mrza and Wang describes the latest developments in the field. Overall, the manuscript presents valuable information regarding iPSC-Derived Microglia. However, there are a few areas that require more detail and clarity, which will improve the quality of the manuscript:

1.     How will the heterogeneity of iPSC-derived microglia (in terms of differentiation and maturity) be addressed?

2.     Do iMGs differentiate into M1 and M2, or into M2a, M2b, and M2c?

3.     iPSC-derived microglia may not fully recapitulate the characteristics of primary microglia isolated from the central nervous system. How is the immaturity of the iMGs addressed?

4.     It is indeed important to address the ethical concerns surrounding iMGs.

5.     It would be beneficial to include subheadings in the text, especially in sections concerning 'Immunized Organoids in Neurodegenerative Diseases' (Section 6).

Round 2

Reviewer 1 Report

Comments and Suggestions for Authors

The majority of the concerns have been sufficiently tackled thus the issues raised or tasks assigned have been addressed adequately.

Comments on the Quality of English Language

There is still potential and room for improvement, yet this does not impede acceptance in its current state.